# OpenReview forum: "LAION-C: An out-of-distribution benchmark for web-scale vision models"
_ICLR.cc/2025/Conference — Submitted to ICLR 2025_

### Official Review · Reviewer_JfPP · 2024-10-29

**Soundness:** 3
**Presentation:** 3
**Contribution:** 4
**Rating:** 6
**Confidence:** 3

**Summary:**

This paper introduces LAION-C, a new out-of-distribution (OOD) dataset specifically designed for models trained on web-scale data, which can sometimes overlap with benchmarks like ImageNet-C. To mitigate this issue, the authors generated a dataset containing six types of image distortions, each across five severity levels. Experiments demonstrate the challenging OOD properties of LAION-C for existing models. Furthermore, the authors conducted psychophysical experiments to assess prediction consistency between human observers and models, revealing that current models can now match or even surpass the performance of the best human observers.

**Strengths:**

1. Clear structure and presentation: The paper is well-organized, with a logical structure and smooth flow that improves readability and comprehension.
2. Insightful motivation for OOD benchmarking: This dataset addresses the critical issue of data leakage in current benchmarks. This approach highlights the limitations of evaluating current models trained on web-scale data.
3. Interesting distortion design: The authors designed six distinct distortion types that minimize overlap with common real-world corruptions, reducing the likelihood of data leakage. This approach improves the dataset’s utility as a truly challenging OOD benchmark.

**Weaknesses:**

1. Limited novel insights in model ranking comparisons: Figure 4 shows minimal divergence in the ranking of model performance between ImageNet-C and LAION-C. Similarly, Figure 3 demonstrates a strong linear correlation between model performance on ImageNet-C and LAION-C. Since a benchmark dataset's purpose is to reveal different performance characteristics across models, more analysis is needed to clarify the unique insights LAION-C provides beyond what ImageNet-C already offers.

**Questions:**

1. Performance differences between CNNs and transformers: Given that LAION-C primarily features patch-based transformations, are there observed differences in how CNNs and transformers perform on these types of distortions?

2. Class disparity between datasets: ImageNet-C includes 1,000 classes, whereas LAION-C has only 16, potentially creating an imbalance in comparison. Excluding the psychophysical experiments, what methods or adjustments could be employed to address this disparity and ensure a fair evaluation between the datasets?

**Details Of Ethics Concerns:**

Human subjects are included in psychophysical experiments.

---

> ### Author Response · Authors · 2024-11-20
> **Response to Reviewer JfPP**
>
> Dear Reviewer, \
> Thank you for your helpful review and interesting questions! We were pleased to read that you found our paper **clearly structured** and believe that we **address a critical issue**. Your main observation regarding the lack of differences in the resulting model rankings provides an interesting perspective. We conducted an investigation of other well-established OOD-benchmark datasets, which revealed a high degree of correlation between all of them, even though they measure different aspects of robustness (see here: https://ibb.co/93k5ns5). Arguably, the ranking of models is not the only relevant conclusion to be drawn from these benchmarks, but the absolute performance of models matters as well. In our case, LAION-C reveals that all models struggle more with our synthetic corruptions than they do with the more reasonable corruptions of ImageNet-C.
>
> *Q: “[A]re there observed differences in how CNNs and transformers perform on these types of distortions?”* \
> *A:* We have created a new figure showing the differences in performance between CNNs and Transformer-based models: https://ibb.co/2dhpqkn. In summary, transformers always perform better than CNNs, but it is important to keep in mind that the latter class contains smaller and older models. Our figure 10 shows a more detailed breakdown of models and implies the same conclusion.
>
> *Q: “What methods or adjustments could be employed to address this disparity [of number of classes] and ensure a fair evaluation between the datasets?”* \
> *A:* Thank you for this interesting question. We have also created a 1000-class version of LAION-C, but to ensure a fair comparison between human and machine performance, we implemented the 16-class version (since the 1000-class version would be practically impossible for human observers). The current intensity levels of the corruptions areis also tuned for such scenarios. Conversely, the 16-class version of ImageNet-C does not pose a significant challenge to our suite of models. We provide an additional figure comparing the different datasets here: https://ibb.co/Z2Mzv3d. The code we will release upon publication will allow other researchers to apply our corruptions to arbitrary images with adjustable strength, so re-creating a 1000-class version of LAION-C is possible, should the need arise.

---

### Official Review · Reviewer_qKi1 · 2024-11-03

**Soundness:** 3
**Presentation:** 3
**Contribution:** 2
**Rating:** 5
**Confidence:** 4

**Summary:**

LAION-C is a new benchmark dataset designed to evaluate the out-of-distribution (OOD) robustness of modern vision models trained on web-scale datasets. ​It includes six distortion types across five severity levels, intended to be OOD even for large datasets like LAION. ​
The paper argues that ImageNet-C is no longer effective for evaluating the robustness of models because many common corruptions in ImageNet-C, such as blur and JPEG compression artefacts, are already present in web-scale datasets. ​It includes a thorough evaluation of nearly 60 models as well as a comparison to human annotators.

**Strengths:**

Comprehensive Evaluation: The paper thoroughly evaluates 58 vision models on the LAION-C dataset, comparing their performance to human observers. This evaluation highlights the challenges posed by LAION-C and the variability in model robustness. ​

Human evaluation: The most insightful outcomes in the paper stem from the comparison to humans, which was done in a controlled setting. This is valuable as it allows good insights into the commonalities and differences between the algorithms and human perception.

**Weaknesses:**

## Performance Correlation with ImageNet-C
The paper motivates LAION-C mainly in comparison with ImageNet-C. As a pure OOD benchmark, it is unclear which advantage the new benchmark brings other than being harder and less saturated. Looking at Figure 3, there is an almost perfect alignment between ImageNet-C and LAION-C performance: models that perform better on ImageNet-C also perform better on LAION-C with only minimal changes between very similarly performing models. With this, the question is if the same study could have been performed using ImageNet-C instead?

## Nature of the distribution shifts
The paper proposes six new synthetic distortions. The motivation behind synthetic distortions is two-fold: on the one hand, they are easy to apply at a large scale. On the other hand, they are less likely to be contained in web-scale datasets. The experiments show what is intuitively clear: this data is further away from the training distribution of large models than ImageNet-C. The question, however, is whether these distribution shifts are _useful_.
Potentially useful shifts come in two types, which I will call A and B:
* Type-A is the most straightforward: actual real-world distribution shifts that allow understanding of the model’s performance on in-the-wild data.
* Type-B shifts are synthetic and might not appear in the real world, but they can be motivated by human/animal visual systems being robust in response to these changes. This allows for studying the differences between biological and artificial perception.

The shifts presented here are not of type A. It is difficult to imagine many real-world scenarios resulting in the proposed image corruption. They are also not of type B: for example, Figure 5 shows that humans are not very good at understanding these corruptions.
What does the proposed OOD dataset tell us about the benchmarked models? It is not real-world performance, and it is also not how aligned they are with biological perception. What remains is the insight that the proposed distortions affect humans more than the models, which is somewhat interesting but not particularly useful.

**Questions:**

> 19 human subjects are briefly presented with a distorted image and are asked to classify it into one of 16 classes, reminiscent of how a DNN might be evaluated on a classification task.

How is this process reminiscent of the forward pass in a network? As a field, we are responsible for not overclaiming the “intelligence” or the relatedness of models to biological intelligence.

**Details Of Ethics Concerns:**

The paper includes a study with human participants and thus should include an ethics statement, potentially an ethics committee approval, and a discussion of the compensation of the participants.

As a dataset/benchmark paper, it should also include a datasheet or other dataset report.

---

> ### Author Response · Authors · 2024-11-20
> **Response to Reviewer qKi1**
>
> Dear Reviewer, \
> Thank you for your insightful review. We were delighted to read that you **valued our human experiments and the comprehensiveness of our model evaluations.**
>
> *Q: “[Could] the same study [...] have been performed using ImageNet-C instead?”* \
> *A:* Thank you for bringing up this point. The crucial difference between ImageNet-C and our new LAION-C dataset is that the ImageNet-C corruptions (such as noise, frost and blur) are fairly natural and appear in today’s large-scale training sets, thus they cannot be used for OOD testing anymore. In contrast, we measure OOD-performance on a highly synthetic, but truly OOD dataset. The fact that (the effectively in-distribution) performance on ImageNet-C still seems to be a good predictor of OOD performance is arguably nontrivial, but in line with work such as [1]. See also our response to reviewer wQbV for an analysis of the correlation between LAION-C and well-established OOD benchmarks.
>
> *Q: “[Are] these distribution shifts [...] useful?”* \
> *A:* Using your nomenclature, LAION-C would indeed be a dataset of type B: fully synthetic, and allowing us to study the differences between human and machine vision. Specifically, the difference is that models pick up on features that seem inaccessible to human observers, allowing them to solve even the harder intensity levels of our corruptions. This insight seems useful to us, since DNNs have been proposed as models of human vision [2,3], but one would want a good model to exhibit similar behavior (including failure modes) as the modeled system. We think that datasets consisting of corruptions to which the primate visual system is robust are not the only useful kind of dataset, but that there also is value in showing inconsistencies in “the other direction”, i.e. datasets where models are robust, but humans are not.
> Going beyond that, there is another purpose that LAION-C serves: Since it is so synthetic and unrealistic (i.e., not type A) we can really use it to measure the OOD robustness of models trained on web-scale datasets, which essentially have to be assumed to contain everything that is realistic, rendering truly OOD type A datasets practically impossible.
>
> *Q: “How is this process reminiscent of the forward pass in a network?”* \
> *A:* Thank you for the question, we have now clarified that section of the paper. We were not referring to the information processing in participants’ visual cortex, but only to the task itself: Human participants are presented with an image and perform a classification task in which they assign one of 16 possible classes to each image, which is similar to how supervised models are evaluated (vs. for example an open response format, in which they could provide an arbitrary textual answer).
>
> [1] Accuracy on the Line: On the Strong Correlation Between Out-of-Distribution and In-Distribution Generalization https://arxiv.org/abs/2107.04649 \
> [2] Deep neural networks: a new framework for modeling biological vision and brain information processing https://www.annualreviews.org/content/journals/10.1146/annurev-vision-082114-035447 \
> [3] Brain-Like Object Recognition with High-Performing Shallow Recurrent ANNs https://arxiv.org/abs/1909.06161

---

> > ### Comment · Reviewer_qKi1 · 2024-11-27
> >
> > Thank you for the responses!
> >
> > Interestingly, my first question was in the opposite direction of reviewer wQbV: my question was (and is partially confirmed by the correlation experiment) if this dataset can give us new insights about a model that other datasets cannot. A high positive correlation in performance likely means that models that perform well on one dataset also perform well on the other.
> >
> > Regarding the second question, I am happy with the argument that one could use the dataset to study the difference between human and model perception, as there seem to be clear differences. However, the second part is unclear to me: if there is no true OOD anymore because of web-scale training, the value of studying artificial true OOD diminishes.

---

> > > ### Author Response · Authors · 2024-11-29
> > > **Response-2 to qKi1**
> > >
> > > Dear Reviewer qKi1,\
> > > Thank you for your insightful comments, we appreciate your acknowledgment of our dataset's role in studying the differences between human and model perception.\
> > > We understand your question was whether our dataset can provide new insights about models that other datasets cannot. The correlation analysis we conducted for reviewer wQbV was done mainly to establish that high correlation is common among OOD benchmarks testing different distribution shifts, indicating that such correlations are normal and not indicative of a lack of unique insights. \
> > > To clarify the second point: We do not mean to suggest that true OOD no longer exists. Instead, we suggest that Type A OOD scenarios are highly unlikely given the breadth of web-scale datasets. Therefore, studying artificial true OOD, as provided by LAION-C, remains valuable. It serves as a proxy for unseen challenges, allowing us to probe the limits of current models' robustness in a controlled and challenging environment.\
> > > Furthermore, we have updated our manuscript to include an explicit ethics statement and a datasheet for LAION-C as you requested. We hope our response clarifies the unique contributions and importance of the LAION-C dataset.

---

### Official Review · Reviewer_wQbV · 2024-11-05

**Soundness:** 3
**Presentation:** 3
**Contribution:** 2
**Rating:** 6
**Confidence:** 4

**Summary:**

In this era of web scale vision models where distortion from ImageNet-C could be part of their training data, the improvements on ImageNet-C is not strongly correlated to model robustness. This paper proposes a new corruption benchmark to evaluate model’s robustness against distribution shifts. Authors carefully craft six synthetic new corruptions with five different severity levels that are visually different from ImageNet-C and real world data distribution. Authors evaluate different variety of models and also examine human performance on this benchmark. Results suggest that models struggle to perform on this benchmark when compared to ImageNet-C.

**Strengths:**

+ Presentation of problem formulation is clear. Figure 1 provide qualitative evidence that distortions of ImageNet-C are part of training data LAION 400M.

+ The proposed distortions are disjoint from ImageNet-C.

+ A wide range of vision models are evaluated against this benchmark to show its effectiveness.

+ Highly appreciate the experiments with human subjects to ensure comparability.

**Weaknesses:**

It is not clear on how to realize that improvement on the proposed distortions would translate to the improved generalization of the model. How do we know that these distortions would act as proxy to determine model generalization? Does improving on these distortions imply improving model performance on diverse unseen real world distribution shifts? To address this issue, it is important to establish that LAION-C correlates with generalization to real-world distribution shifts. Authors can evaluate different model performances on existing real-world OOD datasets and analyze their correlation with LAION-C performance.

As acknowledged by the authors, certain distortions disrupt the object features beyond visual recognition even at severity level 3 (e.g. Stickers and Geometric Shapes). It is not clear how the parameters for these severity levels are chosen. As some of the distortions mainly occlude the object of interest in the image, a ratio of occlusion of the object under these distortions would be helpful to understand the severity level. Besides, it would be informative to provide quantitative metrics of each severity level across all distortion types.

The argument that LAION-C is challenging but solved by finetuning is not convincing as the model improvements on the extreme severities could have been the result from learning spurious or shortcut features, and does not correspond to learning object features. Authors could examine this behaviour by analyzing model's attention patterns before and after fine-tuning, and test the fine-tuned model on other OOD datasets to assess genuine robustness improvements.

A prior work [a] introduced an alternative to ImageNet-C dataset called ImageNet-C-bar with distortions dissimilar to different augmentations used in the training, would these distortions eligible to become part of LAION-C?

[a] On Interaction Between Augmentations and Corruptions in Natural Corruption Robustness

https://openreview.net/pdf?id=zbEupOtJFF

https://github.com/facebookresearch/augmentation-corruption

**Questions:**

Please refer weakness section.

------------------------
Final review: I thank the authors for conducting additional analyses. In the light of additional correlation analysis and quantitative measure of corruptions, my concerns are addressed and increase my score to marginal acceptance.

---

> ### Author Response · Authors · 2024-11-20
> **Response to Reviewer wQbV**
>
> Dear Reviewer, \
> Thank you for your insightful comments! We were delighted to read that you found the **presentation of our work clear** and see the **value in the human experiments** we conducted. We highly appreciate your in-depth review of our work and believe that the additional experiments we conducted to address your suggestions have greatly improved the manuscript.
>
> *Q: “Does improving on these distortions imply improving model performance on diverse unseen real world distribution shifts?”* \
> *A:* That’s a great question. Did you have any specific dataset in mind for unseen real world distribution shifts? To address your question, we analyzed the correlations between model performance on LAION-C and on several established OOD datasets, including IN-C, IN-A, IN-R and IN-Sketch. Specifically, we subsampled a total of 18 models trained on both LAION-2B and ImageNet (IN) to control for architectural variations, focusing on recent architectures such as ViT and ConvNext-inspired models. Our analysis reveals a positive correlation between model performance on LAION-C and all investigated OOD datasets, suggesting that performance gains on LAION-C do indeed translate to other distribution shifts. We have updated our findings in section 3.3 , and you can also find the results here: https://ibb.co/93k5ns5.
>
> *Q: “A ratio of occlusion of the object under these distortions would be helpful to understand the severity levels”* \
> *A:* Thanks for the suggestion! We have updated section A.4 to include the occlusion ratio of the objects for geometric shapes and stickers (you can also access the table here: https://ibb.co/bJjnV7p).  In the same section, we also explain in more detail how we constructed each distortion. More generally, we share your intuition: At higher intensity levels, most of our corruptions are increasingly hard to solve. In principle, we could have chosen to keep only one intensity level (e.g. level 2) or limited our analysis to, say, the first four levels only. However, it is a common practice in psychophysics to test the full range of difficulties, from perfect performance to chance performance (e.g., see [1] Figure 2). ImageNet-C falls short in this regard, since even the hardest severity can often be solved too easily. We decided to include the harder images, leaving the choice of their inclusion in other experiments to the researchers conducting those. We also wanted to leave room for future models that potentially drastically improve upon this reference model, which simply did not happen for any of the evaluated models.
>
> *Q: “Fine-tuning experiment: [I]mprovements on the extreme severities could have been the result from learning spurious or shortcut features”* \
> *A:* Thank you for raising this point. We fully agree that high model performance on the hardest severity levels is likely a consequence of models learning strategies that differ greatly from those that humans would use. It seems that we haven’t sufficiently explained the goal of the fine-tuning experiment: The purpose is not to suggest that fine-tuning improves robustness, but simply to quantify that even at high levels of noise there is signal left—as opposed to purely random noise, in which case fine-tuning wouldn’t be able to improve over chance performance. There is still some learnable signal that allows correct classification, even if it is inaccessible to humans (i.e., “spurious features”). We think of the performance after fine-tuning as an upper bound that serves only to put the performance of other models and humans into perspective. We have now updated section 3.4 to better explain this idea.
>
> *Q: “[W]ould [the ImageNet-C-bar] distortions be eligible to become part of LAION-C?”* \
> *A:* LAION-C could indeed be extended by further distortions such as ImageNet-C-bar, provided that the authors agree and that the distortions don’t appear in a large dataset like LAION already. We appreciate the pointer to this related work as it further emphasizes the importance of perceptually dissimilar corruptions for OOD benchmarking scenarios. We have updated our manuscript to include this work in our related work section.
>
> [1] Partial success in closing the gap between human and machine vision https://openreview.net/forum?id=QkljT4mrfs

---

### Official Review · Reviewer_jFJK · 2024-11-06

**Soundness:** 3
**Presentation:** 3
**Contribution:** 2
**Rating:** 5
**Confidence:** 3

**Summary:**

During the ImageNet era, researchers developed numerous benchmarks to test how vision models perform on out-of-distribution data, with ImageNet-C emerging as one of the most widely used benchmakrs. ImageNet-C featured four types of corruptions: noise, blur, weather, and digital effects. However, the paper points out a key limitation: these corruptions are commonly found in web-scale training data, which makes them less effective for measuring true out-of-distribution robustness for web-scale models. To address this shortcoming, they've developed a new benchmark called LAION-C. This benchmark introduces six new, manually designed corruption types: Mosaic, Glitched, Vertical Lines, Stickers, Geometric Shapes, and Checkerboard patterns.

**Strengths:**

When it comes to benchmarking robust image classification, this paper presents a comprehensive and thorough set of experiments. The addition of psychophysics experiments is also valuable.

**Weaknesses:**

- Why is there no evaluation of Vision-Language models?
- The paper, as it currently stands, doesn't adequately justify why we need another robustness benchmark for image classification, especially since it reaches similar conclusions to [1]. While the paper claims their benchmark is more challenging than ImageNet-C, this alone isn't particularly compelling. During the ImageNet era, ImageNet-C served as a good benchmark for testing out-of-distribution performance. However, now that we're in the age of Vision-Language foundation models, shouldn't we be going beyond simple image classification? Shouldn't we also be testing these models' ability to reason and explain their classification decisions for out-of-distribution samples?

**Questions:**

- What was your reasoning for selecting these 6 specific types of corruption?
- Some of the corruption levels, particularly at intensities 4 and 5, seem extremely severe. Looking at Figure 2, even a human would struggle to identify the images with Mosaic at intensities 4 and 5, Stickers at 3, 4, and 5, and Geometric Shapes at 3, 4, and 5. What's the rationale for including these extremely challenging cases? What meaningful insights can we gain from seeing models perform poorly on images that are essentially unrecognizable? If even humans can't identify these images, what value does testing models on them provide?

---

> ### Author Response · Authors · 2024-11-20
> **Response to Reviewer jFJK**
>
> Dear Reviewer, \
> Thank you for your constructive suggestions and insightful questions. We greatly appreciate that you found our **experiments valuable and thorough**. We have updated the manuscript to better motivate why another robustness benchmark might indeed be helpful, see section 3.3 – in short, we believe that existing benchmarks no longer suffice, since they contain corruptions that are also present in today’s web-scale datasets. Could you provide the reference [1] you were referring to, so we can take a look?
>
> *Q: “Why is there no evaluation of Vision-Language models?”*\
> *A:* Thanks for this excellent suggestion. We have now evaluated GPT-4o on a representative subset (48k samples) of LAION-C images and updated figures 3, 4 and 10 accordingly (see also here: https://ibb.co/album/wYwg7y). This analysis shows no fundamental differences between VLMs and the other models we evaluated, but this is interesting in itself and we appreciate the suggestion. Since we provide LAION-C as a publicly available dataset,the community will be able to test future models as well.
>
> *Q: “What was your reasoning for selecting these 6 specific types of corruption?”* \
> *A:* Our 6 specific types of corruption have been selected from a larger set of candidate corruptions on the grounds of eliciting the greatest variance in performance of a suite of models, while also being parameterized and cheap enough to compute for a large dataset. For the purpose of a “truly OOD” test set, the specific choice of corruption types matters less than the fact that they were not present in the training data distribution, a finding that we extensively quantify in Section 3.1 of the paper. We now explain this motivation more clearly in Section 2.1.
>
> *Q: “What's the rationale for including these extremely challenging cases?”* \
> *A:* It is correct that at higher intensity levels, most of our corruptions are increasingly hard to solve. In principle, we could have limited our analysis to, say, the first four levels only. However, it is a common practice in psychophysics to test the full range of difficulties, from perfect performance to chance performance (e.g., see [1] Figure 2). ImageNet-C falls short in this regard, since even the hardest severity can often be solved with very high accuracies, limiting its use as a challenging benchmark. We also wanted to leave room for future models that potentially drastically improve upon this. That said, no model is expected to perform well on the hardest severity level, which we now state clearly in Section 2.1.
>
> *Q: “If even humans can't identify these images, what value does testing models on them provide?”* \
> *A:* For many applications of AI (autonomous driving, medical imaging), super-human performance is desirable. As models get better, benchmarks need to become more challenging. For instance, when the popular SWE-Bench coding benchmark was introduced, the best model performed at only 4% (and many untrained humans would do even worse), while now models are at nearly 50% performance. Since humans have, historically,  been the gold standard for OOD-performance (as e.g. stated in [1,2]), our experiment aims to assess how the current generation of models performs on difficult stimuli compared to humans. We find that modern models now outperform humans on OOD tasks, while earlier models still struggle to do so (see appendix A.6). However, the low error consistency between humans and models implies that contemporary models do not achieve  "super-human" performance by merely replicating human visual strategies. Instead, models are leveraging unique cues and strategies beyond those used by humans.
>
> [1] Partial success in closing the gap between human and machine vision https://openreview.net/forum?id=QkljT4mrfs \
> [2] Human and DNN classification performance on images with quality distortions: A comparative study https://dl.acm.org/doi/10.1145/3306241

---

> > ### Comment · Reviewer_jFJK · 2024-11-27
> >
> > Thank you for your responses, which clarifies many of my questions. I apologize for missing the actual reference for [1]. The paper I meant to refer to is [1] Geirhos et al. "Partial success in closing the gap between human and machine vision".
> >
> > It's great to see results from VLMs like GPT-4o for image classification, but my main point was about going beyond just simple image classification. The key distinction between VLMs and traditional image classification models lies in their ability to explain their reasoning behind predictions, perform in-context learning / more powerful reasoning at inference time. For instance, if we allocate more computational resources during inference, VLMs might eventually address most out-of-distribution challenges in image classification. I believe providing early evidence for this kind of hypothesis using datasets like LAION-C could offer significant value to the community, rather than focusing on simple image classification.

---

> > > ### Author Response · Authors · 2024-11-29
> > > **Response-2 to jFJK**
> > >
> > > Dear Reviewer,\
> > > Thank you for your thoughtful comments and for clarifying your main point regarding VLMs. We fully agree that exploring the reasoning and in-context learning capabilities of VLMs is an exciting and valuable direction for research.\
> > > To address your suggestion and demonstrate how LAION-C could be used to probe VLMs, we conducted an initial experiment to investigate how in-context samples help VLMs in out-of-distribution challenges. Specifically, we prompt VLMs to classify LAION-C images, but provide N labelled in-context samples ($N \in [1,3,5]$) with the same corruption. We then evaluate the model on 100 images per class under these conditions. The results can be seen here: https://ibb.co/YD3YfxG. \
> > > For GPT-4o, we find that in-context learning yields modest, but stable performance gains: Classification accuracy increases from ~44% with no context to ~47% with 5 context samples, suggesting that providing labelled examples from the new distribution is a viable method of improving OOD-classification performance at inference time. \
> > > The overall increase of Gemini’s performance from context size 0 to 5 is similar, but we observe an initial dip for context size 1, which suggests that limited contexts may introduce noise, temporarily hindering performance. Notably, the responses in this condition were not biased towards the label of the in-context sample, as one might have guessed. For much larger context sizes, we did not observe further performance gain.\
> > > Additionally, we have conducted another experiment inspired by your comments to investigate the visual reasoning capabilities of Gemini and GPT, by asking the models to provide not only a response, but also a justification of their decision. When the models arrive at the correct solution, they typically report the presence of appropriate features. But both models also fail on images that still seem fairly easy to human observers and confidently hallucinate nonsensical features, in line with the low error consistency values we measured in our psychophysical experiment. Find examples for GPT here: https://ibb.co/Bz6w2Zq and for Gemini here: https://ibb.co/12hP52x. \
> > > While we believe these preliminary results provide valuable insights, the primary focus of our paper remains the establishment of the LAION-C dataset. By introducing this challenging benchmark, we aim to provide a valuable resource to enable studies like this one to further explore and understand the advanced capabilities of VLMs. Thanks again for suggesting that “providing early evidence for this kind of hypothesis using datasets like LAION-C could offer significant value to the community”.

---

### Author Response · Authors · 2024-11-20
**General Response**

We would like to thank all reviewers for taking the time to review our paper so thoroughly and providing such constructive and helpful feedback. We are delighted to read that reviewers believe we “**address [a] critical issue**” (JfPP) and conduct a “**comprehensive and thorough set of experiments**” (jFJK) that “**allows good insights into the commonalities and differences between the algorithms and human perception.**” (qKi1)

We incorporated the following changes into the next revision of the manuscript to address reviewer suggestions:

- Multiple reviewers (wQbV, qKi1) questioned the utility of LAION-C as a proxy for real-world distribution shifts, which is a valid concern, since our corruptions are highly synthetic. To address these concerns, we have conducted an additional experiment (see section 3.3) demonstrating that performance gains on LAION-C indeed translate to performance gains on real-world OOD distributions. That said, the core idea of the paper is not to measure real-world performance, but to measure a model’s ability to generalize beyond its training data, which requires a truly OOD test set. LAION-C fulfills this requirement, while ImageNet-C does not.

- Several reviewers (jFJK, wQbV) wondered about how the severity levels for our corruptions were chosen, and why we chose to include the hardest levels, even though they seem nearly impossible. We have updated our manuscript to explain the procedure in more detail, see sections 2.1 and A.4 – in short, we employed a common practice from psychophysics, which is to cover the full range of corruption strengths all the way from perfect performance to chance level performance. Models are not expected to perform above chance at the hardest level, which we now state explicitly.

- We evaluated GPT-4o on a large subset of LAION-C samples and included the results in figures 3, 4 and 10 (find a quick preview of figures here: https://ibb.co/album/wYwg7y)
- As requested, we calculated the object occlusion ratio for the Sticker and Geometric Shapes corruptions as an additional quantitative measurement of the distortions, see section A.4 (you can also access the table here: https://ibb.co/bJjnV7p).
- We conducted a correlation analysis of model performance on LAION-C and a selection of established OOD datasets, to answer whether improving on LAION-C implies improved model performance on diverse unseen real world distribution shifts, find a table of correlations between OOD test sets here: https://ibb.co/93k5ns5.

---

### Author Response · Authors · 2024-11-25
**General Response-2**

Dear Reviewers,\
Thank you again for your helpful feedback. We have further expanded our analysis to include another state-of-the-art VLM, Gemini 1.5 Pro, which we evaluated on large subsets of both LAION-C and ImageNet-C (here: https://ibb.co/album/N6fksk; thanks to reviewer jFJK  for the idea). Additionally, we have calculated the Error Consistency (EC) between human observers and GPT-4o / Gemini, just like we did for the other models. We find relatively large differences in EC between the two VLMs, which you can see here: https://ibb.co/kgfWhd0. We would greatly appreciate your thoughts regarding the revisions to our manuscript.

---

### Meta-Review · Area_Chair_BE9P · 2024-12-21

**Metareview:**

This paper proposes an out-of-distribution dataset for benchmarking models for "web-scale'' models. The authors explained that distribution shifts, such as additive noise or weather corruptions, in existing datasets, are seen by the web-scale models. In other words, these datasets no longer serve as a good OOD benchmark.  The reviewers found that the paper's experiment is comprehensive, and the writing is overall clear. However, reviewers are concerned about the contribution and why the proposed dataset is useful. Critically, the dataset synthetically generates corruption such as Mosaic, vertical lines... etc. While it is true that these added corruptions are OOD for the web-scale models, why are these the right corruptions for evaluating OOD? Considering ImageNet-C, the added corruption such as weather, blur, and noise are somewhat ``naturally occurring'', however, this is not the case for the proposed LAION-C. The authors did respond that the LAION-C serves as a proxy for other existing datasets, in this case, what is the additional contribution brought by the dataset, as pointed out by reviewer qKi1.

**Additional Comments On Reviewer Discussion:**

Many requested details and concerns were addressed during the discussion period. However, the main issue of the importance of the dataset, and concerns why we should study these synthetically generated corruption remain unresolved. Two of the reviewers were not entirely convinced after the discussion period. The AC agrees with the reviewer and finds the motivation/explanation for choosing the corruptions, in Section 2.1, to be insufficient.

---

### Decision · Program_Chairs · 2025-01-22

Reject